# Transcriptional Up-Regulation of FBXW7 by K_Ca_1.1 K^+^ Channel Inhibition through the Nrf2 Signaling Pathway in Human Prostate Cancer LNCaP Cell Spheroid Model

**DOI:** 10.3390/ijms25116019

**Published:** 2024-05-30

**Authors:** Susumu Ohya, Hiroaki Kito, Junko Kajikuri, Yohei Yamaguchi, Miki Matsui

**Affiliations:** Department of Pharmacology, Graduate School of Medical Sciences, Nagoya City University, Nagoya 467-8601, Japan; kito@med.nagoya-cu.ac.jp (H.K.); kajikuri@med.nagoya-cu.ac.jp (J.K.); y_yamagu@med.nagoya-cu.ac.jp (Y.Y.); miki.matsui.4238@gmail.com (M.M.)

**Keywords:** cancer stemness, spheroid, Ca^2+^-activated K^+^ channel, K_Ca_1.1, Nrf2, FBXW7, protein degradation, C/EBP, miR223, c-Myc

## Abstract

The tumor suppressor gene F-box and WD repeat domain-containing (FBXW) 7 reduces cancer stemness properties by promoting the protein degradation of pluripotent stem cell markers. We recently demonstrated the transcriptional repression of FBXW7 by the three-dimensional (3D) spheroid formation of several cancer cells. In the present study, we found that the transcriptional activity of FBXW7 was promoted by the inhibition of the Ca^2+^-activated K^+^ channel, K_Ca_1.1, in a 3D spheroid model of human prostate cancer LNCaP cells through the Akt-Nrf2 signaling pathway. The transcriptional activity of FBXW7 was reduced by the siRNA-mediated inhibition of the CCAAT-enhancer-binding protein C/EBP δ (CEBPD) after the transfection of miR223 mimics in the LNCaP spheroid model, suggesting the transcriptional regulation of FBXW7 through the Akt-Nrf2-CEBPD-miR223 transcriptional axis in the LNCaP spheroid model. Furthermore, the K_Ca_1.1 inhibition-induced activation of FBXW7 reduced (1) K_Ca_1.1 activity and protein levels in the plasma membrane and (2) the protein level of the cancer stem cell (CSC) markers, c-Myc, which is a molecule degraded by FBXW7, in the LNCaP spheroid model, indicating that K_Ca_1.1 inhibition-induced FBXW7 activation suppressed CSC conversion in K_Ca_1.1-positive cancer cells.

## 1. Introduction

The large-conductance Ca^2+^-activated K^+^ channel, K_Ca_1.1, encoded by the KCNMA1 gene is activated by membrane depolarization and by an increase in intracellular Ca^2+^ concentration ([Ca^2+^]_i_) [1]. In non-excitable cells, including cancer cells, K_Ca_1.1 activation-induced hyperpolarization increases the electrochemical gradient for Ca^2+^ through voltage-independent, store-operated Ca^2+^ channels, resulting in increases in the [Ca^2+^]_i_, and K_Ca_1.1 activation also reduces [K^+^]_i_, resulting in the modulation of intracellular signaling [1,2]. K_Ca_1.1 plays a critical role in the proliferation, migration, apoptosis, and metastasis of cancer cells by controlling Ca^2+^ signaling and cell volume [3]. The overexpression of K_Ca_1.1 correlates with tumorigenicity, a higher grade, invasiveness, poor prognosis, and stemness in cancer [1,4]. Recent studies suggest a relationship between K_Ca_1.1 and resistance to chemotherapies in cancer spheroid models with stem cell-like characteristics [5,6].

The tumor suppressor gene F-box and WD repeat domain-containing (FBXW) 7 is one of the most commonly known E3 ubiquitin ligases in cancers [7]. The loss of FBXW7 is strongly associated with the maintenance of cancer cell stemness because it contributes to the protein degradation of cancer stemness markers, such as c-Myc and Nanog [8,9]. In prostate cancer, up-regulated FBXW7 also suppresses the expressions of c-Myc and Notch-1 [10]. In addition, FBXW7 plays an important role in the modification of sensitivity to anti-cancer drugs in cancers, including prostate cancer [11]. We previously identified FBXW7 as an essential regulator of K_Ca_1.1 protein degradation in several cancers [5,6,12] and showed that the down-regulated expression of FBXW7 by cancer spheroid formation increased K_Ca_1.1 activity by promoting its protein level in the plasma membrane [5,6]. FBXW7 is down-regulated by oncogenic signaling pathways such as Akt in cancer cells [7,13].

CCAAT/enhancer binding proteins (C/EBPs) consist of six isoforms: C/EBPα (CEBPA), β (CEBPB), γ (CEBPG), δ (CEBPD), ε (CEBPE), and ζ (CEBPZ, CHOP) [14]. Previous studies reported that CEBPD negatively regulated FBXW7 gene expression in cancer cells and was essential for the acquisition of cancer stemness properties [15,16]. In prostate cancer, CEBPD is involved in cancer stemness, metastasis, and resistance to anti-cancer drugs [17].

In Hermonizome 3.0 (https://maayanlab.cloud/Hermonizome on 20 November 2023), FBXW7 is one of the CEBPD target genes (10705) in chromatin immunoprecipitation sequencing (ChIP-seq) datasets taken from the ENCODE Transcription Factor Targets database. In addition, the knockdown of C/EBPs decreases Nanog gene expression by reducing the activity of its promoter [18].

Oncogenic microRNAs (miRNAs) promote cancer development by targeting tumor suppressor genes [19], and miRNAs such as miR223 and miR135a play an essential role in the function of FBXW7 in cancer cells [7]. miR223 is mostly known as a target for FBXW7 and negatively correlates with FBXW7 expression in cancer cells [20]. It is also a potential biomarker for prostate cancer diagnosis [21]. The miR223/FBXW7 axis contributes to the acquisition of resistance to anti-cancer drugs, such as doxorubicin (DOX), and the promotion of epithelial–mesenchymal transition and metastasis in cancer cells [22,23,24]. In Hermonizome 3.0, miR-223 and miR135a are both included as CEBPD target genes in ChIP-seq datasets.

The present study aimed to investigate the K_Ca_1.1 inhibition-mediated regulation of FBXW7 expression and its associated signaling cascades, including CEBPs and miRNAs, in K_Ca_1.1-expressing cancer cells. The obtained results suggest that the down-regulation of FBXW7 by 3D spheroid formation was reversed by the inhibition of K_Ca_1.1 through the Akt-Nrf2-CEBPD-miR223 transcriptional axis.

## 2. Results

### 2.1. FBXW7 Is Up-Regulated by K_Ca_1.1 K^+^ Channel Inhibition in the LNCaP Spheroid Model

Previous studies [5,6] showed that the down-regulated expression of FBXW7 by 3D spheroid formation suppressed the protein degradation of K_Ca_1.1 in human prostate cancer LNCaP cells, which increased K_Ca_1.1 activity. We herein examined the effects of treatment with the K_Ca_1.1 inhibitor, PAX, on FBXW7 expression and its associated signaling cascades in the LNCaP spheroid model by real-time PCR and Western blot assays. Consistent with previous findings [5], the expression of FBXW7 transcripts was markedly reduced by the 3D spheroid formation of LNCaP cells (*n* = 4, *p* < 0.01) (Figure 1A). In the present study, FBXW7 was significantly up-regulated by the treatment with PAX for 12 h in the LNCaP spheroid model (*n* = 4, *p* < 0.01) (Figure 1B). Accordingly, the protein level of FBXW7 with a molecular weight of approx. 70 kDa was significantly increased by the PAX treatment for 48 h in the LNCaP spheroid model (*n* = 4, *p* < 0.05) (Figure 1C,D). These results suggest that the long-term inhibition of K_Ca_1.1 reduced K_Ca_1.1 activity by promoting its protein degradation and reversed cancer stemness by enhancing the FBXW7-mediated protein degradation of pluripotent marker proteins in the LNCaP spheroid model.

### 2.2. K_Ca_1.1 Inhibition-Induced Up-Regulation of FBXW7 Is Mediated through the Akt-Nrf2 Signaling Pathway in the LNCaP Spheroid Model

We recently reported that the inhibition of K_Ca_1.1 suppressed the Akt-Nrf2 signaling pathway by reducing the phosphorylation level of Akt in the LNCaP spheroid model [2]. Wang et al. indicated that activated Akt down-regulated FBXW7 in cancer cells [11]. To elucidate whether the K_Ca_1.1 inhibition-induced up-regulation of FBXW7 is mediated through the Akt/Nrf2 signaling pathway, we examined the effects of the Akt activator, SC79, and the Nrf2 activator, NK252, on it in the LNCaP spheroid model. Co-treatment with SC79 (10 μM) and PAX (10 μM) for 12 h reversed the up-regulation of FBXW7 in the LNCaP spheroid model (*n* = 4, *p* < 0.01 vs. +/−) (Figure 1E). No significant changes in the relative expression level of FBXW7 transcripts were found using a single treatment with SC79 in the LNCaP spheroid model: 0.935 ± 0.031 (*n* = 4, *p* > 0.05 vs. −/−). Similarly, co-treatment with NK252 (100 μM) and PAX (10 μM) for 12 h significantly reversed the up-regulation of FBXW7 (*n* = 4, *p* < 0.01 vs. +/−) (Figure 1F). No significant changes in the relative expression level of FBXW7 transcripts were noted by a single treatment with NK252 in the LNCaP spheroid model: 0.947 ± 0.076 (*n* = 4, *p* > 0.05 vs. −/−).

### 2.3. K_Ca_1.1 Inhibition Reduces the Nuclear Translocation of Phosphorylated Nrf2 (P-Nrf2) in Isolated Cells from the LNCaP Spheroid Model

To investigate whether the inhibition of K_Ca_1.1 reduces the nuclear translocation of P-Nrf2 in isolated cells from LNCaP spheroids, the cellular localization of P-Nrf2 was visualized 2 h after the treatment with 10 μM PAX by laser-scanning confocal fluorescence microscopy. The anti-P-Nrf2 antibody and nuclei were labeled with the Alexa Fluor 488-conjugated secondary antibody and DAPI, respectively (Figure 2A,B). P-Nrf2 expression was assessed as the percentage of cells with P-Nrf2-positive nuclei. The percentage of cells with P-Nrf2-positive nuclei was significantly reduced by the treatment with 10 μM PAX (*n* = 6, *p* < 0.01) (Figure 2C). Unphosphorylated Nrf2 (Nrf2) staining was detected in the cytosolic regions of vehicle- (Figure 2D) and PAX-treated (Figure 2E) LNCaP cells at similar levels. Approximately 90% of the vehicle control in LNCaP cells showed P-Nrf2-positive nuclei (Appendix A), and no significant changes in the percentage of cells with P-Nrf2-positive nuclei were found by the treatments with SC79 and NK252 (Appendix A).

### 2.4. Involvement of CEBPD in the K_Ca_1.1 Inhibition-Induced Up-Regulation of FBXW7 in the LNCaP Spheroid Model

Balamurugan et al. reported the involvement of the transcriptional regulator CEBPD in the inhibition of FBXW7 expression in cancer cells [15]. As shown in Figure 3A, the expression level of CEBPD was increased by the 3D spheroid formation of LNCaP cells. In a previous study, CEBPB was also up-regulated together with 3D spheroid formation [2]. Therefore, we examined the effects of the siRNA-mediated inhibition of CEBPB and CEBPD on the expression level of FBXW7 transcripts. The inhibition of CEBPD up-regulated FBXW7 in the LNCaP spheroid model (*n* = 4, *p* < 0.01) (Figure 3B), whereas no significant change was observed in FBXW7 levels following the inhibition of CEBPB (*n* = 4, *p* > 0.05). We then investigated the effects of the inhibition of K_Ca_1.1 with PAX on the expression level of CEBPD transcripts in the LNCaP spheroid model. The expression level of CEBPD was significantly reduced by a treatment with 10 μM PAX for 12 h in the LNCaP spheroid model (*n* = 4, *p* < 0.01) (Figure 3C). We also investigated the effects of the siRNA-mediated inhibition of Nrf2 on the expression level of CEBPD transcripts in the LNCaP spheroid model. The expression level of CEBPD was significantly reduced by the inhibition of Nrf2 in the LNCaP spheroid model (*n* = 4, *p* < 0.01) (Figure 3D). CEBPB, CEBPD, and Nrf2 expression levels were selectively inhibited by siRNA transfection, with an inhibition efficacy of less than 50% (Appendix A). These results strongly suggest that CEBPD was responsible for the transcriptional repression of FBXW7 by 3D spheroid formation through the Nrf2 signaling pathway and indicate its potential as a crucial target for the K_Ca_1.1 inhibition-induced up-regulation of FBXW7.

### 2.5. Involvement of microRNA(s) in the K_Ca_1.1 Inhibition-Induced Up-Regulation of FBXW7 in the LNCaP Spheroid Model

We hypothesized that miRNAs are involved in the CEBPD inhibition-mediated up-regulation of FBXW7 in the LNCaP spheroid model. Chu et al. showed increases in miR135a levels in CEBPD-overexpressing astrocytoma cells [25]. In addition, Liu et al. indicated that the expression level of FBXW7 transcripts was decreased by the overexpression of miR223 [20]. As shown in Figure 4A,B, the expression level of miR223 but not miR135a was significantly increased by 3D spheroid formation. The expression level of miR223 was significantly reduced by the treatment with 10 μM PAX for 12 h in the LNCaP spheroid model (*n* = 4, *p* < 0.01) (Figure 4D), whereas that of miR135a was not (*p* > 0.05) (Figure 4C). Also, the treatment with the Nrf2 inhibitor, ML385 (10 μM), for 12 h significantly reduced the expression level of miR223 (*n* = 4, *p* < 0.01) but not miR135a (*p* > 0.05) (Figure 4E,F). Additionally, the results from experiments using miRNA mimics or inhibitors revealed that the expression levels of FBXW7 transcripts were higher in the miR223 inhibitor (miR223-i)-transfected LNCaP spheroid model than in the miRNA inhibitor, a negative control (Cont-i)-transfected model (*n* = 4, *p* < 0.01) (Figure 4G). No significant change was noted in FBXW7 levels in the miR135a inhibitor (miR135a-i)-transfected LNCaP spheroid model (*n* = 4, *p* > 0.05 vs. Cont-i). On the other hand, the expression levels of FBXW7 transcripts were lower in the miR223 mimic (miR223-m)-transfected LNCaP spheroid model than in the miRNA mimic, a negative control (Cont-m)-transfected model (Figure 4H). No significant changes were found in the FBXW7 levels in the miR135a mimic (miR135a-m)-transfected LNCaP spheroid model. These results strongly suggest that the K_Ca_1.1 inhibition-induced up-regulation of FBXW7 was mediated by the Akt/Nrf2/CEBPD/miR223 axis.

### 2.6. K_Ca_1.1 Inhibition-Induced Down-Regulation of Nanog Is Mediated by Its Transcriptional Repression through the Akt-Nrf2 Signaling Pathway in the LNCaP Spheroid Model

Nanog is a possible substrate of FBXW7-mediated protein degradation and leads to chemoresistance in cancer [8,9]. Multidrug resistance proteins (MRPs), which are involved in the acquisition of DOX resistance, were up-regulated together with the spheroid formation of LNCaP cells through the Akt-Nrf2-Nanog axis [2,5]. Similarly, previous studies indicated the impact of Akt-Nrf2-Nanog-mediated MRPs on chemoresistance and cancer stemness. Consistent with our previous findings [2,5], the expression levels of Nanog transcripts were significantly reduced by the PAX treatment and the siRNA-mediated inhibition of Nrf2 (Figure 5A,B). However, siRNA-mediated CEBPD inhibition and miR223 inhibitor did not affect Nanog transcription (Figure 5C,D), suggesting that the K_Ca_1.1 inhibition-induced down-regulation of MRP5 overcame DOX resistance mainly by the transcriptional repression of Nanog through the Akt-Nrf2 signaling pathway but not by the increased protein degradation of Nanog through the CEBPD-miR223-FBXW7 axis.

### 2.7. K_Ca_1.1 Inhibition-Induced FBXW7 Up-Regulation Reduces the Activity of K_Ca_1.1 by Promoting Its Protein Degradation in the LNCaP Spheroid Model

Based on the results obtained thus far, we hypothesized that the inhibition of K_Ca_1.1 promotes the FBXW7-mediated protein degradation of K_Ca_1.1 in the LNCaP spheroid model, increasing both its protein expression in the plasma membrane and its activity. To assess the protein expression level of K_Ca_1.1 in the plasma membrane, we performed immunocytochemical staining with an Alexa Fluor 488-conjugated anti-K_Ca_1.1 antibody, which distinguishes the extracellular region of K_Ca_1.1 in fixed, non-permeabilized cells of the LNCaP spheroid model. We successfully visualized K_Ca_1.1 in the plasma membrane of isolated cells from the LNCaP spheroid model using confocal imaging (Figure 6A). We then examined the effects of the treatment with PAX for 48 h on the protein expression levels of K_Ca_1.1 in the plasma membrane by flow cytometric (Figure 6B–D) and Western blotting (Figure 6E,F) analyses. The obtained results showed that the protein expression levels of K_Ca_1.1 were significantly decreased by the treatment with PAX (*p* < 0.01, *n* = 5 and 4 in Figure 4D,F, respectively). No significant change was noted in the expression level of K_Ca_1.1 transcripts using the treatment with PAX (*n* = 4 for each, *p* > 0.05) (Figure 6G). These results suggest that K_Ca_1.1 activity was reduced by long-term K_Ca_1.1 inhibition mediating FBXW7 in the LNCaP spheroid model.

We then compared 1 μM PAX-sensitive outward K^+^ currents elicited between vehicle- and PAX-pretreated (for 48 h) cells of the LNCaP spheroid model by depolarizing voltage steps between −80 and +60 mV from a holding potential of −60 mV using whole-cell patch clamp recordings (Figure 7). Before current measurements, isolated cells were incubated with normal culture media in which PAX was excluded for more than 4 h. Outward currents were almost completely blocked by the application of 1 μM PAX in both groups (Figure 7A,B). PAX-sensitive current density at +60 mV was significantly larger in the vehicle-pretreated group (*n* = 13) than in the PAX-pretreated group (*n* = 15, *p* < 0.01) (Figure 7C,D). Therefore, K_Ca_1.1 activity was reduced by the long-term exposure to K_Ca_1.1 inhibitors in the LNCaP spheroid model, followed by FBXW7-mediated protein degradation.

### 2.8. K_Ca_1.1 Inhibition-Induced FBXW7 Up-Regulation Reduces the Protein Expression Level of the Cancer Stemness Marker, c-Myc, in the LNCaP Spheroid Model

One of the markers for cancer stemness, c-Myc is a well-known major target for protein degradation by FBXW7, and the down-regulated expression of FBXW7 is associated with the accumulation of c-Myc proteins in cancer cells [26]. The expression levels of c-Myc transcripts were similar between 2D monolayers and 3D spheroids of LNCaP cells (*n* = 4, *p* > 0.05) (Figure 8A); however, the expression levels of c-Myc proteins were significantly higher in 3D spheroids than in 2D monolayers (*n* = 4, *p* < 0.01) (Figure 8B,C). We next examined the effects of the inhibition of K_Ca_1.1 by PAX on c-Myc protein expression in the LNCaP spheroid model. In contrast to FBXW7 (see Figure 1C,D), the protein level of c-Myc, with a molecular weight of approx. 50 kDa, was significantly decreased by the PAX treatment in the LNCaP spheroid model (*n* = 4, *p* < 0.01) (Figure 8D,E), without changes in the transcriptional expression level of c-Myc by the treatment with PAX for 12 h (*n* = 4, *p* > 0.05) (Figure 8F). In addition, no significant changes were observed in the protein level of c-Myc in the miR223 inhibitor (miR223-i)-transfected LNCaP spheroid model (*n* = 4, *p* > 0.05 vs. Cont-i) (Figure 8G). These results suggest that the K_Ca_1.1 inhibition-induced up-regulation of FBXW7 inhibited cancer stemness by destabilizing c-Myc proteins in the LNCaP spheroid model.

## 3. Discussion

The loss of the E3 ubiquitin ligase, FBXW7 synergizes with the acquisition of stemness and anti-cancer resistance and the accumulation of pluripotent transcription factors, such as c-Myc in cancer cells [8,9,11]. We recently reported that the inhibition of K_Ca_1.1 overcame chemoresistance by down-regulating drug-metabolizing enzymes through the Akt-Nrf2 signaling pathway in 3D cancer spheroid models [2]. In addition, decreases in FBXW7 have been shown to suppress the protein degradation of K_Ca_1.1, resulting in enhanced K_Ca_1.1 activity in cancer spheroid cells [5,6]. Previous studies demonstrated that FBXW7 was down-regulated through Akt signaling pathways in cancer cells [7,13]; however, the intracellular signal transduction cascades that regulate FBXW7 transcription in 3D cancer spheroids with cancer stemness properties currently remain unclear.

In the present study, we elucidated the molecular mechanisms underlying K_Ca_1.1 inhibition-induced FBXW7 modifications in the human prostate cancer LNCaP spheroid model. The main results obtained are as follows. (1) The inhibition of K_Ca_1.1 promoted the up-regulation of FBXW7 through the Akt-Nrf2-CEBPD-miR223 transcriptional axis (Figure 1, Figure 2, Figure 3 and Figure 4). (2) K_Ca_1.1 inhibition-induced increases in FBXW7 reduced K_Ca_1.1 activity by inhibiting its protein expression (Figure 6 and Figure 7). (3) K_Ca_1.1 inhibition reduced the expression of the core marker of cancer stemness, c-Myc, possibly by the transcriptional up-regulation of FBXW7 (Figure 8).

The transcriptional factor, CEBPD, plays a crucial role in tumor development, metastasis, and resistance to therapies, as well as being distinguished as a driver of maintaining cancer stemness [16]. The present study showed the K_Ca_1.1 inhibition-induced down-regulation of CEBPD in the LNCaP spheroid model as well as a significant increase in its expression by the 3D spheroid formation (Figure 3C). Hartl et al. recently reported the co-amplification of CEBPD and MYC in 25–85% of cancer patients (i.e., uterine carcinoma, hepatocellular carcinoma, breast invasive carcinoma, and prostate adenocarcinoma), but not that of CEBPB and MYC (0%) [16]. Chan et al. indicated that CEBPD prevented the FBXW7-mediated protein degradation of MYC in urothelial carcinoma [27]. However, the mechanisms underlying the MYC/CEBPD relationship have not yet been confirmed. Among pluripotency genes that are known as cancer stem cell (CSC) markers, c-Myc is a substrate of FBXW7-mediated protein degradation [8,9]. In the present study, the K_Ca_1.1 inhibition-induced up-regulation of FBXW7 decreased the protein expression level of c-Myc in the LNCaP spheroid model, without changing the expression level of c-Myc transcripts (Figure 8D–F). To date, several compounds and molecules suppressing CEBPD expression have been developed [16,28]. Therefore, K^+^ channel inhibitors may be useful as novel agents that suppress the CEBPD pathway and may reverse c-Myc-mediated cancer stemness by up-regulating FBXW7.

The inhibition of K_Ca_1.1 overcame DOX resistance by repressing CYP3A4 transcription through the Akt-Nrf2-CEBPB axis in the LNCaP spheroid model [2]. It has been reported that resistance to anti-cancer drugs is promoted through the Nrf2 signaling pathway in prostate cancer cells [29]. The up-regulated expression of CEBPB by 3D spheroid formation was suppressed by the inhibition of K_Ca_1.1 in the LNCaP spheroid model, and Nrf2 and CEBPB cooperatively contributed to drug resistance [2]. In the present study, siRNA-mediated CEBPB inhibition did not elicit the up-regulation of FBXW7 in the LNCaP spheroid model (Figure 3B), which is consistent with the findings reported by Balamurugan et al. (2010), who showed that CEBPB did not affect FBXW7 promoter expression in cancer cells [15]. Conversely, the up-regulated expression of CEBPD by 3D spheroid formation was suppressed by the inhibition of K_Ca_1.1 in the LNCaP spheroid model; however, neither CEBPD inhibition nor miR223 inhibition changed the expression levels of CYP3A4 transcripts in the LNCaP spheroid model (Appendix A). These results strongly suggest that CEBPD is an independent upstream regulator of FBXW7 gene expression in the LNCaP spheroid model. A previous study reported that the expression of CYP3A4 was regulated by c-Myc in the liver [30]. However, the present results suggest that the K_Ca_1.1 inhibition-induced down-regulation of CYP3A4 overcame DOX resistance by the transcriptional repression of CEBPB through the Akt-Nrf2 signaling pathway, but not by the increased protein degradation of c-Myc through the CEBPD-miR223-FBXW7 axis.

miRNAs are associated with the pathogenesis of various types of human malignancies, including prostate cancer, and affect post-transcriptional modifications of molecules, which are attributed to the stemness features of CSCs [31,32]. The following miRNAs act as negative regulators of FBXW7 transcription: miR25, miR32, miR92b, miR96, miR155, miR182, miR223, and miR367 [19]. Among these miRNAs, miR223 is up-regulated in several 3D spheroids expressing pluripotency genes [33]. In addition, miR223 possesses broad regulatory roles for FBXW7 in many cancers [19]. In the present study, post-transcriptional modifications to FBXW7 through miR223 were responsible for the regulation of K_Ca_1.1 activity in the LNCaP spheroid model (Figure 4G,H). Recent studies indicated the potential of miRNAs including miR223 as potential therapeutic targets for various diseases, such as K^+^-channel-related cardiovascular disorders (voltage-gated K^+^ channel, K_V_4.2/KCND2) and epilepsy (voltage-gated K^+^ channels, K_V_2.1/KCNMB1 and K_V_7.2/KCNQ2) [34,35]. The present study demonstrated that miR223 inhibitors may increase the expression of FBXW7, ultimately promoting the protein degradation of K_Ca_1.1 and suppressing K_Ca_1.1 activity in K_Ca_1.1-expressing CSCs. Therefore, miR223 mimics/enhancers/inhibitors have the potential to be K_Ca_1.1-modulating agents/molecules. To elucidate whether K_Ca_1.1 inhibition-induced increases in FBXW7 promoted the protein degradation of K_Ca_1.1, further experiments, such as a protein degradation assay and ubiquitylation detection assay will be needed.

We previously reported that K_Ca_1.1 transcription was activated by the inhibition of the Akt/mTOR signaling pathway in breast cancer MDA-MB-453 cells [12]. Correspondingly, in the LNCaP spheroid model, treatment with the Akt inhibitor, AZD5363 (2 μM) for 12 h activated K_Ca_1.1 transcription by approx. 25% (Appendix A). Sun et al. indicated that the activation of Nrf2 increased K_Ca_1.1 protein levels through transcriptional activation in arterial smooth muscle cells [36]. However, no significant changes were observed in the expression level of K_Ca_1.1 transcripts by the treatment with the Nrf2 inhibitor, ML385 (10 μM), and the Nrf2 activator, NK252 (100 μM), for 12 h and the transfection of siNrf2 in the LNCaP spheroid model (Appendix A). In addition, the phosphorylation of FBXW7 at T205 through ERK1/2 activation induced the protein degradation of FBXW7 in cancer cells [37,38]. Zhang et al. showed that silencing K_Ca_1.1 decreased ERK1/2 phosphorylation in non-cancerous cells [39]. Therefore, ERK signaling may affect K_Ca_1.1 inhibition-induced increases in FBXW7 protein levels in the LNCaP spheroid model (see Figure 1B–F). However, no signals were detected in P-ERK1/2 levels in vehicle- and PAX-treated LNCaP spheroids (Appendix A). These results suggest that transcriptional modifications to K_Ca_1.1 through the Nrf2 and ERK signaling pathways are not responsible for the regulatory mechanism of K_Ca_1.1 function in the LNCaP spheroid model.

## 4. Materials and Methods

### 4.1. Chemicals and Reagents

The following chemicals and reagents were used: paxilline (PAX) and AZD5363 from Cayman Chemical (Ann Arbor, MI, USA); RPMI 1640 culture medium, Trypsin/EDTA solution, and 5-α-dihydrotestosterone (DHT) from FUJIFILM Wako Pure Chemical (Tokyo, Japan); fetal bovine serum (FBS) and 4′,6-diamino-2-phenylindole (DAPI) from Sigma-Aldrich (St. Louis, MO, USA); ML385 from Selleckchem (Huston, TX, USA); donkey anti-rabbit IgG H&L (Alexa Fluor^®^ 488) from Abcam (Bristol, UK); Silencer^®^ Select Pre-designed siRNAs as negative control No. 1, human CEBPB (s2892), human CEBPD (s2895), human Nrf2 (NFE2L2, s9491), and human FBXW7 (s224356) from Life Technologies Japan (Tokyo, Japan); anti-FBXW7 polyclonal (rabbit), anti-Nrf2 polyclonal (rabbit), anti-P-Nrf2 (S40) monoclonal (rabbit), and ERK1/2 polyclonal (rabbit) antibodies from ABclonal (Tokyo, Japan); anti-K_Ca_1.1 (APC-151, extracellular) polyclonal (rabbit) antibody from Alomone Labs (Jerusalem, Israel); anti-P-ERK1/2 (T202/Y204) monoclonal (rabbit) antibody from R&D Systems (Minneapolis, MN, USA); anti-c-Myc polyclonal (rabbit) antibody from ProteinTech (Rosemont, IL, USA); anti-ACTB monoclonal (mouse) antibody from Medical & Biological Laboratories (Nagoya, Japan); anti-rabbit and mouse horseradish peroxidase-conjugated IgG antibodies from Merck Millipore (Darmstadt, Germany); SC79, NK252, hsa-miR135a miRNA mimic, hsa-miR135a-5p miRNA inhibitor, hsa-miR223 miRNA mimic, hsa-miR223-5p miRNA inhibitor, miRNA mimic Negative control #1, and miRNA inhibitor Negative control #1 from MedChemExpress (Monmouth Junction, NJ, USA); ReverTra Ace from ToYoBo (Osaka, Japan); flat-bottomed dishes and plates from Corning (Corning, NY, USA); PrimeSurface 96U plates from Sumitomo Bakelite (Tokyo, Japan); Luna Universal qPCR Master Mix from New England Biolabs Japan (Tokyo, Japan); Lipofectamine RNAiMAX Transfection Reagent and ECL Western Blotting Detection Reagent from Thermo Fisher Scientific (Waltham, MA, USA); CytoFix/Perm kit from BD Pharmingen (Franklin Lakes, NJ, USA); glass bottomed dishes from Matsunami Glass (Osaka, Japan); FastGene RNA Premium kit and FastGene miRNA Enhancer kit from Nippongenetics (Tokyo, Japan); Mir-X miRNA First-Strand Synthesis kit from TaKaRa (Osaka, Japan); PCR primers from Nihon Gene Research Laboratories (Sendai, Japan). The other chemicals used in the present study were from Sigma-Aldrich, FUJIFILM Wako Pure Chemical, and Nacalai Tesque (Kyoto, Japan), unless otherwise stated.

### 4.2. Cell Culture

The human prostate cancer cell line, LNCaP, was purchased from the RIKEN Cell Bank (Osaka, Japan) and cultured in RPMI 1640 medium, supplemented with 10% FBS, penicillin (100 units/mL)–streptomycin (100 μg/mL), and 5 nM DHT [5,6] under a humidified atmosphere containing 5% CO_2_ at 37 °C. Flat-bottomed dishes and plates were used in the two-dimensional (2D) cell culture. Regarding 3D spheroid formation, cell suspensions were seeded onto PrimeSurface 96U plates at 10^4^ cells/well and then cultured for 7 days. For cell dissociation or cell detachment, trypsin (0.25 *w*/*v*%)/EDTA (1 mM) solution was used.

### 4.3. RNA Extraction, cDNA Synthesis, and Real-Time PCR

Total RNA was extracted from cells using the conventional acid guanidium thiocyanate–phenol–chloroform method. Reverse transcription was performed using ReverTra Ace with random hexanucleotides. miRNA was extracted using a FastGene RNA Premium kit and FastGene miRNA Enhancer kit, and cDNA synthesis was performed by Mir-X miRNA First-Strand Synthesis kit, according to the manufacturer’s protocol. Quantitative real-time PCR was conducted using the Luna Universal qPCR Master Mix and the Applied Biosystems 7500 Fast Real-Time PCR system (Thermo Fisher Scientific) [5,6]. PCR primers of human origin are listed in Appendix A [40]. Relative expression levels were calculated using the 2^−ΔΔCt^ method and normalized to ACTB or U6.

### 4.4. Western Blots

Collected spheroids were washed with phosphate buffer solution, and then protein lysates were extracted using radioimmunoprecipitation assay buffer. Equal amounts of protein were subjected to SDS-PAGE and immunoblotting with anti-FBXW7 polyclonal (rabbit) (1:2000) (approx. 70 kDa), anti-K_Ca_1.1 polyclonal (rabbit) (1:750) (approx. 100 kDa), anti-c-Myc polyclonal (rabbit) (1:4000) (approx. 50 kDa), P-ERK1/2 monoclonal (rabbit) (1:1500) (approx. 40 kDa), ERK1/2 polyclonal (rabbit) (1:3000) (approx. 40 kDa), and anti-ACTB monoclonal (mouse) (1:15,000) (approx. 45 kDa) antibodies, and were then incubated with anti-rabbit or -mouse IgG horseradish peroxidase-conjugated antibodies. An ECL Western Blotting Detection Reagent was used to detect the bound antibody. The resulting images were analyzed using Amersham Imager 600 (GE Healthcare Japan, Tokyo, Japan). The optical density of the protein band signal relative to that of the ACTB signal was calculated using ImageJ (Fiji distribution) software, and protein expression levels in the vehicle control were then expressed as 1.0.

### 4.5. Confocal Imaging of K_Ca_1.1 in the Plasma Membrane and the Nuclear Translocation of P-Nrf2

Isolated cells from the 3D spheroid model of LNCaP cells were treated with 10 μM PAX for 48 h and fixed with 2% paraformaldehyde buffer. Non-permeabilized cells were stained with an anti-K_Ca_1.1 (extracellular) polyclonal antibody followed by an Alexa Fluor 488-conjugated secondary antibody. Besides that, isolated cells were treated with 10 μM PAX for 2 h and then fixed and permeabilized using the CytoFix/Perm kit. Anti-P-Nrf2 and anti-Nrf2 antibodies were labeled with an Alexa Fluor 488-conjugated secondary antibody, and nuclei were labeled with DAPI. Fluorescence images were visualized using a confocal laser scanning microscope system (Nikon A1R, Tokyo, Japan) [41]. Cells stained by an Alexa Fluor 488-conjugated anti-K_Ca_1.1 antibody were subjected to an analysis on the BD FACSCanto II Flow Cytometry system (BD Biosciences, San Jose, CA, USA), acquiring at least 10,000 events per sample [42].

### 4.6. Transfection with siRNAs, miRNA Mimics, and miRNA Inhibitors

Lipofectamine RNAiMAX reagent (Thermo Fisher Scientific) was used in the siRNA-mediated inhibition of CEBP isoforms and Nrf2, the miRNA mimic-mediated activation of miR135a and miR223, and miRNA inhibitor-mediated inhibition, according to the manufacturer’s protocol. Silencer Select pre-designed siRNAs for the negative control (siCont), CEBPB (siCEBPB), CEBPD (siCEBPD), and Nrf2 (siNrf2) and pre-designed miRNAs for the negative control for the miRNA mimic (Cont-m), miR135a miRNA mimic (miR135a-m), miR223 miRNA mimic (miR223-m), the negative control for miRNA inhibitor (Cont-i), miR135a-5p miRNA inhibitor (miR135a-i), and miR223-5p miRNA inhibitor (miR223-i) were transfected into adherent monolayer cells. All small nucleic acids were used at a final concentration of 10 nM. Twenty-four hours later, transfected cells were seeded onto PrimeSurface 96U plates, and then cultured for an additional 7 days. The expression levels of the target transcripts were assessed using real-time PCR.

### 4.7. Measurements of K_Ca_1.1 Activity by Whole-Cell Patch Clamp Recordings

A whole-cell patch clamp was applied to 3D-cultured LNCaP cells pre-incubated with/without 10 μM PAX for 48 h using the Axon Patch Clamp System (Axopatch 200B amplifier, Molecular Devices, San Jose, CA, USA) at room temperature (23 ± 1 °C) [5]. Data acquisition and analyses of whole-cell currents were performed using Axon Digidata 1550B plus HumSilencer and Axon pCLAMP software v11.1. (Molecular Devices). Whole-cell currents were measured in the voltage-clamp mode and induced by 500 ms voltage steps, every 15 s, from −80 to +60 mV at a holding potential of −60 mV. The external solution was as follows (in mM): 137 NaCl, 5.9 KCl, 2.2 CaCl_2_, 1.2 MgCl_2_, 14 glucose, and 10 HEPES, pH7.4. The pipette solution was as follows (in mM): 140 KCl, 4 MgCl_2_, 3.2 CaCl_2_, 5 EGTA, 10 HEPES, and 2 Na_2_ATP (pH 7.2, pCa 6.5).

### 4.8. Statistical Analysis

Statistical analyses were performed using the Statistical software XLSTAT (version 2013.1). To assess the significance of differences between two groups and among multiple groups; unpaired/paired Student’s *t*-tests with Welch’s correction or Tukey’s test were used. Results with a *p* value of <0.05 were considered to be significant. Data are presented as means ± the standard error of the mean (SEM).

## 5. Conclusions

CSCs play a critical role in cancer progression and recurrence and, thus, have potential as a therapeutic target for cancer therapy [43]. The present study is the first to support the notion that K^+^ channel inhibition-mediated intracellular signaling may reduce cancer stemness by transcriptionally up-regulating the E3 ubiquitin ligase, FBXW7, in cancer spheroids with CSC properties. The present results provide a valuable in vitro background for future in vivo studies on the K^+^-channel-mediated regulatory mechanism maintaining cancer stemness. With a progressive understanding and ongoing investigations of the potential roles of K^+^ channels in CSCs, K^+^-channel-targeting cancer therapy may decrease recurrence and chemoresistance by eliminating CSCs.

## Figures and Tables

**Figure 1 ijms-25-06019-f001:**
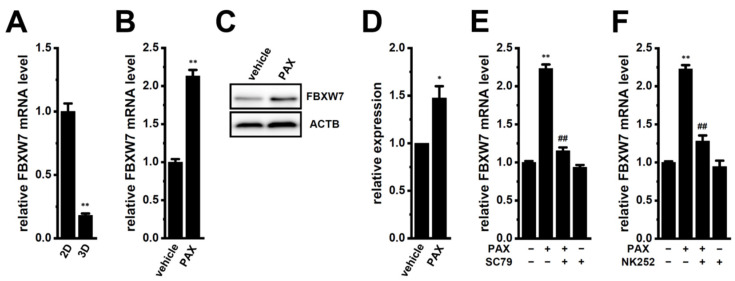
Effects of the treatment with PAX on FBXW7 expression and effects of Akt and Nrf2 activation on the K_Ca_1.1 inhibition-induced up-regulation of FBXW7 in the 3D spheroid model of LNCaP cells. (**A**): Real−time PCR of FBXW7 in 2D monolayers and 3D spheroids of LNCaP cells. After normalization to ACTB mRNA expression levels, FBXW7 mRNA expression levels in 2D-cultured groups were expressed as 1.0 (*n* = 4). (**B**): Real-time PCR of FBXW7 in the LNCaP spheroid models treated with vehicle and 10 μM PAX for 12 h. After normalization to ACTB mRNA expression levels, FBXW7 mRNA expression levels in the vehicle control (vehicle) were expressed as 1.0 (*n* = 4). (**C**): Protein expression of FBXW7 in protein lysates of the LNCaP spheroid models treated with vehicle and 10 μM PAX for 48 h. Blots were probed with anti-FBXW7 (approx. 70 kDa) and anti-ACTB (approx. 45 kDa) antibodies. (**D**): Summarized results were obtained as the optical densities of FBXW7 from (**C**) in the LNCaP spheroid model. After normalization of optical densities of protein band signals with that of the ACTB signal, the optical density in the vehicle control (vehicle) was expressed as 1.0 (*n* = 4). (**E**): Real-time PCR of FBXW7 in the LNCaP spheroid model treated (+) or untreated (−) with 10 μM PAX and 10 μM SC79 for 12 h (*n* = 4). (**F**): Real-time PCR of FBXW7 in the LNCaP spheroid model treated (+) or untreated (−) with 10 μM PAX and 100 μM NK252 for 12 h (*n* = 4). *, **: *p* < 0.05, 0.01 vs. the 2D (**A**), vehicle control (**B**,**D**), and −/− (**E**,**F**) groups. ^##^: *p* < 0.01 vs. the +/− group.

**Figure 2 ijms-25-06019-f002:**
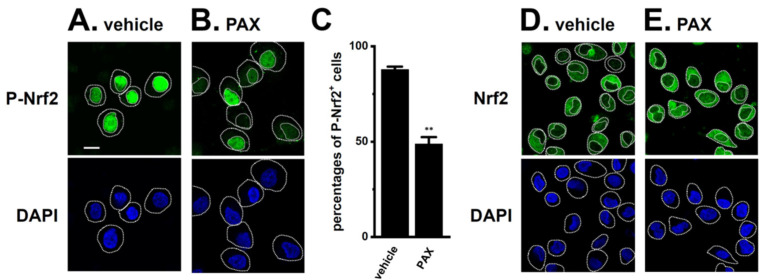
Effects of the PAX treatment on the nuclear translocation of P-Nrf2 in isolated cells from the LNCaP spheroid model. (**A**,**B**): Confocal fluorescent images of Alexa Fluor 488-labeled P-Nrf2 (green, upper panels) in vehicle- (**A**) and PAX (**B**)-treated LNCaP cells. Nuclear morphologies were assessed using DAPI staining (blue, lower panels). Dashed lines show nuclear and cell boundaries. The white bar in (**A**) is calibration (10 μm). (**C**): Summarized results of the percentage of P-Nrf2-positive cells in LNCaP cells (*n* = 6, more than 30 cells for each data point). **: *p* < 0.01 vs. the vehicle control. (**D**,**E**): Confocal fluorescent images of Alexa Fluor 488-labeled Nrf2 (green, upper panels) in vehicle- (**D**) and PAX- (**E**) treated LNCaP cells. Nuclear morphologies were assessed using DAPI staining (blue, lower panels).

**Figure 3 ijms-25-06019-f003:**
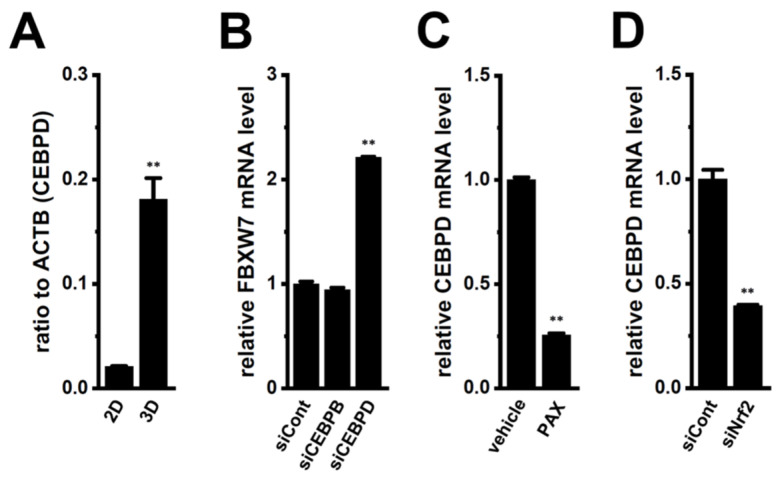
Comparison of expression levels of CEBPD transcripts between 2D monolayers and 3D spheroids of LNCaP cells, effects of the siRNA-mediated inhibition of CEBPB and CEBPD on expression levels of FBXW7 transcripts in the LNCaP spheroid model, and effects of the pharmacological inhibition of K_Ca_1.1 with 10 μM PAX for 12 h and the siRNA-mediated inhibition of Nrf2 on expression levels of CEBPD transcripts in the LNCaP spheroid model. (**A**): Real-time PCR of CEBPD in 2D monolayers and 3D spheroids of LNCaP cells (*n* = 4). Expression levels are shown as a ratio to ACTB. (**B**): Real-time PCR of FBXW7 in the LNCaP spheroid models transfected with negative control siRNA (siCont), CEBPB siRNA (siCEBPB), and CEBPD siRNA (siCEBPD). After normalization to ACTB mRNA expression levels, FBXW7 mRNA expression levels in the siCont-transfected group were expressed as 1.0. (**C**): Real-time PCR of CEBPD in the vehicle- and PAX (10 μM)-treated LNCaP spheroid models for 12 h (*n* = 4). (**D**): Real-time PCR of CEBPD in the control siRNA (siCont)- and Nrf2 siRNA (siNrf2)-transfected LNCaP spheroid models (*n* = 4). After normalization to ACTB mRNA expression levels, CEBPD mRNA expression levels in the vehicle control or siCont group were expressed as 1.0. **: *p* < 0.01 vs. the 2D (A), siCont (**B**,**D**), and vehicle control (**C**) groups.

**Figure 4 ijms-25-06019-f004:**
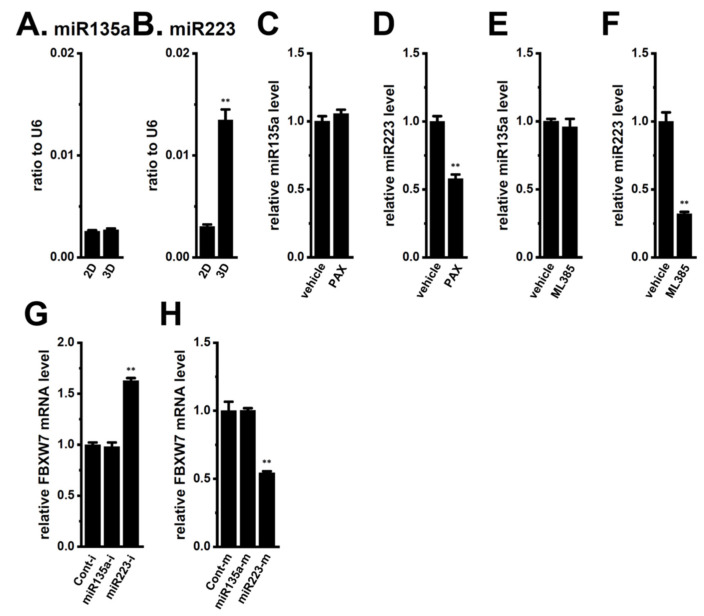
Comparison of expression levels of miR135a and miR223 between 2D monolayers and 3D spheroids of LNCaP cells; the effects of the pharmacological inhibition of K_Ca_1.1 and Nrf2 with 10 μM PAX and 10 μM ML385, respectively, for 12 h on expression levels of miR135a and miR223; and the effects of miR223 modulators on FBXW7 mRNA expression levels in the LNCaP spheroid model. (**A**,**B**): Real-time PCR of miR135a and miR223 in 2D monolayers and 3D spheroids of LNCaP cells (*n* = 4). Expression levels are shown as a ratio to U6. (**C**,**D**): Real-time PCR of miR135a and miR223 in the vehicle- and PAX (10 μM)-treated LNCaP spheroid models for 12 h (*n* = 4). (**E**,**F**): Real-time PCR of miR135a and miR223 in the vehicle- and ML385 (10 μM)-treated LNCaP spheroid models for 12 h (*n* = 4). After normalization to ACTB mRNA expression levels, miR135a and miR223 expression levels in the vehicle controls were expressed as 1.0. (**G**,**H**): Real-time PCR of FBXW7 in the miRNA inhibitor negative control (Cont-i)-, miR-135a-5p miRNA inhibitor (miR-135a-i)-, miR-223-5p miRNA inhibitor (miR-223-i)-, miRNA mimic negative control (Cont-m)-, miR-135a miRNA mimic (miR-135a-m)-, and miR-223 miRNA mimic (miR-223-m)-transfected LNCaP spheroid models (*n* = 4). After normalization to ACTB mRNA expression levels, FBXW7 mRNA expression levels in negative controls (Cont-i and Cont-m) were expressed as 1.0. **: *p* < 0.01 vs. the 2D (**B**), vehicle control (**D**,**F**), Cont-i (**G**), and Cont-m (**H**) groups.

**Figure 5 ijms-25-06019-f005:**
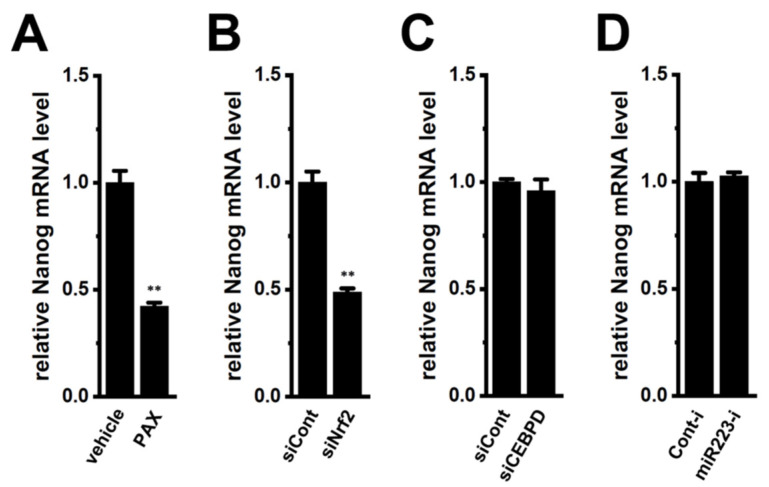
Effects of the pharmacological inhibition of K_Ca_1.1 with PAX for the 12 h, siRNA-mediated inhibition of Nrf2 and CEBPD, and miR223 inhibitor on expression levels of Nanog transcripts in the LNCaP spheroid model. (**A**–**D**): Real-time PCR of Nanog in the LNCaP spheroid models treated with vehicle and PAX (10 μM) for 12 h and transfected with siCont, Nrf2 siRNA (siNrf2), CEBPD siRNA (siCEBPD), miRNA inhibitor negative control (Cont-i), and miR223 inhibitor (miR223-i). After normalization to ACTB mRNA expression levels (*n* = 4), Nanog mRNA levels in the vehicle control, siCont, and Cont-i groups were expressed as 1.0. **: *p* < 0.01 vs. the vehicle control (**A**) and siCont (**B**).

**Figure 6 ijms-25-06019-f006:**
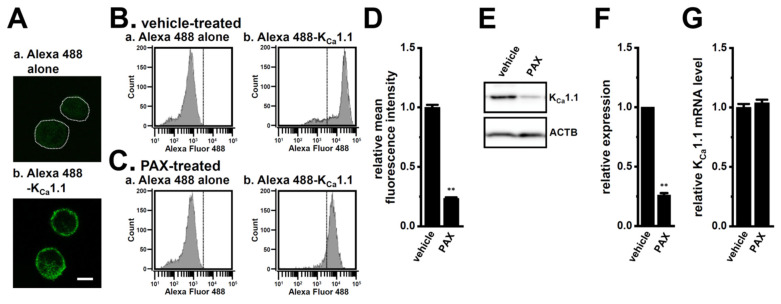
Effects of the treatment with PAX on K_Ca_1.1 expression in the LNCaP spheroid model. (**A**): Immunocytochemical staining of isolated cells from the LNCaP spheroid model for K_Ca_1.1. Fluorescence images of a single confocal plane of the cells labeled with an Alexa Fluor 488 secondary antibody alone (**a**) and Alexa Fluor 488-conjugated K_Ca_1.1 antibody (**b**). In (**a**), dotted lines were used to outline the contours of cells caused by low fluorescence. The white bar in (**b**) is calibration (10 μm). (**B**,**C**): LNCaP spheroids treated with vehicle (**B**) and PAX (10 μM) (**C**) for 48 h were isolated, fixed, stained with an Alexa Fluor 488 alone (**a**) and an Alexa Fluor 488-conjugated K_Ca_1.1 antibody, before being analyzed using flow cytometry. (**D**): Summarized data of the relative mean fluorescence intensity of Alexa Fluor 488. Mean fluorescence intensities in the vehicle control are expressed as 1.0 (*n* = 5). (**E**): The protein expression of K_Ca_1.1 in protein lysates of the LNCaP spheroid model treated with vehicle and 10 μM PAX for 48 h. Blots were probed with anti-K_Ca_1.1 (approx. 100 kDa) and anti-ACTB (approx. 45 kDa) antibodies. (**F**): Summarized results were obtained as the optical densities of K_Ca_1.1 from (**E**) in the LNCaP spheroid model. After normalization of the optical densities of protein band signals with those of the ACTB signal, the optical density in the vehicle control (vehicle) was expressed as 1.0 (*n* = 4). (**G**): Real-time PCR of K_Ca_1.1 in the LNCaP spheroid model treated with vehicle and PAX. After normalization to ACTB mRNA expression levels, K_Ca_1.1 mRNA expression levels in the vehicle control were expressed as 1.0 (*n* = 4). **: *p* < 0.01 vs. the vehicle control.

**Figure 7 ijms-25-06019-f007:**
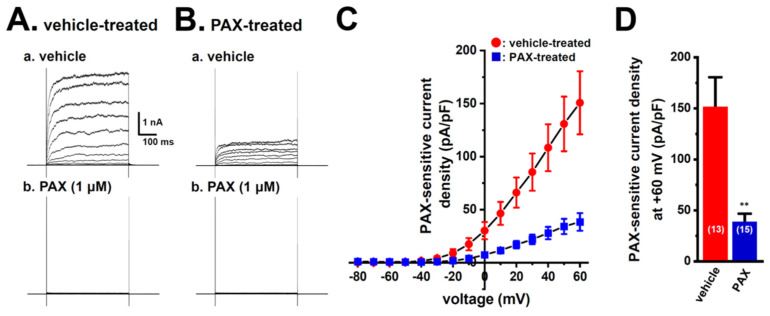
Effects of the treatment with 10 μM PAX for 48 h on PAX-sensitive outward K^+^ currents in isolated cells from the LNCaP spheroid model. (**A**,**B**): Currents were elicited by a 500 ms depolarizing voltage step between −80 and +60 mV from a holding potential (−60 mV), with 10 mV increments in LNCaP cells treated with vehicle (**A**(**a**)) and 10 μM PAX (**B**(**a**)). Before current measurements, isolated cells were incubated with normal culture media in which PAX was excluded for more than 4 h. The application of 1 μM PAX markedly reduced outward K^+^ currents in LNCaP cells (**A**(**b**),**B**(**b**)). (**C**): Current density–voltage relationship for PAX-sensitive peak current density in vehicle-treated (●) and PAX-treated (■) LNCaP cells. (**D**): Summarized results on PAX-sensitive current density at +60 mV in vehicle- and PAX-treated LNCaP cells. Numbers used for experiments are shown in parentheses. **: *p* < 0.01 vs. the vehicle-treated group.

**Figure 8 ijms-25-06019-f008:**
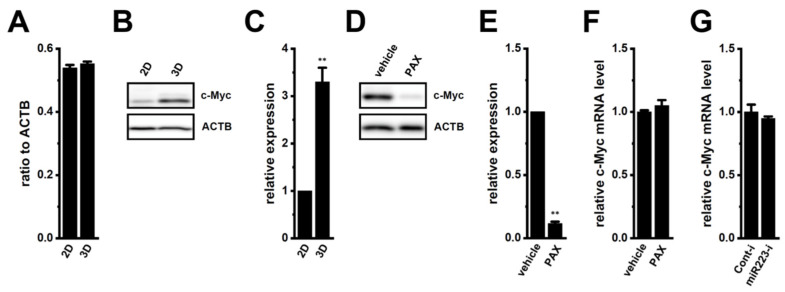
Effects of the treatment with PAX on c-Myc expression in the LNCaP spheroid model. (**A**): Real-time PCR of c-Myc in 2D monolayers and 3D spheroids of LNCaP cells. Expression levels are shown as a ratio to ACTB (*n* = 4). (**B**): Protein expression of c-Myc in protein lysates of 2D monolayers and 3D spheroids of LNCaP cells. Blots were probed with anti-c-Myc (approx. 50 kDa) and anti-ACTB (approx. 45 kDa) antibodies. (**C**): Summarized results were obtained as the optical densities of c-Myc from (**B**). After the normalization of the optical densities of protein band signals with those of the ACTB signal, the optical density in 2D was expressed as 1.0 (*n* = 4). (**D**): Protein expression of c-Myc in protein lysates of the LNCaP spheroid model treated with vehicle and 10 μM PAX for 48 h. Blots were probed with anti-c-Myc and anti-ACTB antibodies. (**E**): Summarized results were obtained as the optical densities of c-Myc from (**D**). Optical density in the vehicle control (vehicle) was expressed as 1.0 (*n* = 4). (**F**): Real-time PCR of c-Myc in LNCaP spheroid models treated with vehicle and 10 μM PAX for 12 h. After normalization to ACTB mRNA expression levels, c-Myc mRNA expression levels in the vehicle control (vehicle) were expressed as 1.0 (*n* = 4). (**G**): Real-time PCR of c-Myc in the miRNA inhibitor negative control (Cont-i)- and miR-223-5p miRNA inhibitor (miR-223-i)-transfected LNCaP spheroid models. After normalization to ACTB mRNA expression levels, c-Myc mRNA expression levels in Cont-i were expressed as 1.0 (*n* = 4). **: *p* < 0.01 vs. the 2D (**C**) and vehicle control (**E**) groups.

## Data Availability

The original contributions presented in the study are included in the article/Appendix A, further inquiries can be directed to the corresponding author.

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
