# Peer review of "Transcriptional Up-Regulation of FBXW7 by K_Ca_1.1 K^+^ Channel Inhibition through the Nrf2 Signaling Pathway in Human Prostate Cancer LNCaP Cell Spheroid Model"

_ijms, 2024, doi:10.3390/ijms25116019_

Round 1

Reviewer 1 Report

Comments and Suggestions for Authors

The authors presented a manuscript describing the fine regulation of Kca1.1 by the AKT/NRF2/CEBPD/Mir223 axis in the LNCaP spheroid model.

The experimental results are obtained through different experimental approaches, including qRT-PCR, western blotting, confocal microscopy, and measurements of KCa1.1 activity by whole-cell patch clamp recordings.

Based on previous studies and the new findings presented in this study, inhibition of Kca1.1 by the inhibitor PAX causes its own degradation by FBXW7, a ubiquitin ligase whose expression is upregulated by the AKT/ axis NRF2/ CEBPD/Mir223 . Overall, the authors provide a further understanding of the complex regulation of Kca1.1, a membrane carrier with high expression in specific tumors.

The experimental design is consistent with the objectives of this study, and the results are mostly convincing; however, there is one point that the authors should consider.

- While the results indicate regulation of FBWX by miR223, there is no direct evidence of PAX effect on endogenous miR223 levels. The authors must provide evidence of the endogenous level of miR223 following treatment with the KCa1.1 inhibitor

- Figure 2 highlights isolated single cells obtained from spheroids. How were these obtained? the authors need to add additional information on how single cells isolated from spheroids are obtained.

Author Response

Responses to Reviewer 1

We would like to thank the reviewer for his/her valuable comments. We have attended to all the points raised by the reviewers. Each comment is highlighted below with our response underneath.

  1. While the results indicate regulation of FBWX by miR223, there is no direct evidence of PAX effect on endogenous miR223 levels. The authors must provide evidence of the endogenous level of miR223 following treatment with the KCa1.1 inhibitor

According to the reviewer’s indication, we added the results of quantitative, real-time PCR examination of miR135a and miR223. As shown in Figure 4, miR223 was upregulated by the 3D spheroid formation, and the expression level was significantly reduced by the treatment with PAX for 12 hr [Page 4, line 154-160; Page 7, line 282-299; Page 12, line 502-504 (Section 4.3.)].

  1. Figure 2 highlights isolated single cells obtained from spheroids. How were these obtained? the authors need to add additional information on how single cells isolated from spheroids are obtained.

In accordance with the reviewer’s suggestion, we added the information on the single cell isolation in the ‘Materials and Methods’ section [Section 4.4. (Page 12, line 511-512)].

Reviewer 2 Report

Comments and Suggestions for Authors

Reviewer Report:

In this manuscript, the authors tackle to understand the downstream mediators of KCa1.1 ion channel regulation through the FBXW7-Nrf2 axis. The concept of understanding the interactions between signaling molecules is indeed intriguing. However, the complexity of the topic is also significant, making the understanding of results challenging. Here, authors sometimes make heavy speculations when summarizing the results, which should be omitted based on the presented evidence. Besides, the manuscript needs some clarification and supporting information to strengthen the claims. Overall, this study merits publication if the necessary adjustments have been made.

Specific comments:

-          The weakest point of the manuscript is that the title generalizes to prostate cancer, yet the research is conducted singly on the LNCaP cell line. It is impossible to interpret the results: Are the observed phenomena only valid on this one cell line, or are they true for androgen-sensitive prostate cancer, or for prostate cancer in general? Would it be different in normal prostate cells? All these questions remain unclear when using only one cell line, making the title also misleadingly broad, as one cannot generalize to prostate cancer with only one cell line. I suggest repeating a few key experiments in at least one other prostate cancer cell line or otherwise the authors. If not possible, the title and appropriate sections should be rewritten.

-          When introducing the role of FBXW7, it might be helpful to specify the types of cancers where its role has been observed. This could provide more context for readers unfamiliar with the specific literature. Also, Have any of the key players in this study been assessed specifically for prostate cancer? To increase relevance, please focus the introduction and discussion of the study more on prostate cancer.

-          The result that „FBXW7 was significantly transactivated by the treatment with PAX for 12 hr in the LNCaP spheroid model” is not evident from Fig. 1B. Only the mRNA expression (transcription) is increased. Usually, transactivation is a phenomenon occurring with proteins and is independent of mRNA expression. Or do the authors intend to say transcription rather than transactivation? Please clarify and adjust. This mistake occurs frequently throughout the text.

-          Please indicate in the appropriate Legends that the Western blot results displayed in the Figures are representative of N=4. Also, the Supplement suggests that the replicates be blotted next to each other: please place appropriate legends on the blots in the Supplement for each N.

-          -pERK1/2 and ERK1/2 original blots are missing in the Original Images.

-          It is very suggestive that upon paxilline application ACTB expression is increased, so it is not a suitable housekeeper for this particular study. Recently, ACTB expression has been proposed as a cancer biomarker rather than a „reference gene” as previously used (https://www.ncbi.nlm.nih.gov/pmc/articles/PMC8806805/). Moreover, KCa1.1 interacts with the cytoskeleton so it would be not surprising if paxilline would perturb ACTB expression. Please include another housekeeper, probably one indicative of cell metabolism (e.g., GAPDH) to rule out the possibility that paxilline has effects on the cytoskeletal protein expression.  

-          On a similar note, is ACTB mRNA expression similarly inhomogeneous as its protein expression? Tools are available to assess whether the „housekeeper” is independent from your experimental conditions (e.g., the geNORM tool). If not, ACTB mRNA would also not be a suitable housekeeper gene.

-          The claim „KCa1.1 inhibition-induced increases in FBXW7 reduced KCa1.1 activity by promoting its protein degradation” is not experimentally assessed in this manuscript, so please show evidence, rephrase, or remove this part. No protein degradation assays have been conducted in this study. One can speculate that the observed effects might be due to protein degradation is the discussion.

-          The next claim (repeated in lines 333-334) is also rather speculation and needs rephrasing and/or removing, or experimental validation: „KCa1.1 inhibition-induced increases in FBXW7 reduced the expression of the core marker of cancer stemness, c-Myc”. Fig. 6 only shows that paxilline impairs c-Myc expression. Here, the consequential link to FBXW7 has not been investigated.

-          It is not clear why the Nanog section is not in the main manuscript but in the supplement, even though the authors discuss it heavily in the Discussion. I suggest placing Fig S6 and the corresponding parts into the main text.

-          Also, regarding lines 333-334, it has not been shown that c-Myc is destabilized—it could simply be less transcribed. Is there any experimental evidence to support increased c-Myc protein degradation, e.g., ubiquitination assay?

-          It is not conceivable what the point of the osteosarcoma cell line is in this study. Why use this osteosarcoma cell line specifically? What are the similarities and dissimilarities between prostate cancer and osteosarcoma?  The observed phenomenon that KCa1.1 inhibition increased the transactivation of FBXW7 through the same axis confuses a reader more than it reassures. I suggest removing it as it osteosarcoma is a completely different topic.

-          The Methods section does not describe how spheroid lysates were extracted and how live cells were isolated from the spheroids for confocal imaging.

-          The siRNA application is also unclear. After siRNA transfection, cells were seeded onto the spheroid-making plates. Were there cells incubated to make spheroids for one week? Please clarify the method.

-          Fig. 5C: Please label in color or in the legend which of the data points indicates vehicle and paxilline treatment.

-          Please include primer sequences in the Supplementary material.

-          DHT abbreviation explanation is missing.

Comments on the Quality of English Language

A minor comment is that the gene expression of human genes and RNA should be written in italic.

Author Response

Responses to Reviewer 2

We would like to thank the reviewer for his/her valuable comments. We have attended to all the points raised by the reviewers. Each comment is highlighted below with our response underneath.

  1. The weakest point of the manuscript is that the title generalizes to prostate cancer, yet the research is conducted singly on the LNCaP cell line. It is impossible to interpret the results: Are the observed phenomena only valid on this one cell line, or are they true for androgen-sensitive prostate cancer, or for prostate cancer in general? Would it be different in normal prostate cells? All these questions remain unclear when using only one cell line, making the title also misleadingly broad, as one cannot generalize to prostate cancer with only one cell line. I suggest repeating a few key experiments in at least one other prostate cancer cell line or otherwise the authors. If not possible, the title and appropriate sections should be rewritten.

We appreciate adequate suggestions from the reviewer. In agreement with the reviewer’s suggestions, We amended the title and confirmed the sections.

  1. When introducing the role of FBXW7, it might be helpful to specify the types of cancers where its role has been observed. This could provide more context for readers unfamiliar with the specific literature. Also, Have any of the key players in this study been assessed specifically for prostate cancer? To increase relevance, please focus the introduction and discussion of the study more on prostate cancer.

We appreciate the reviewer’s suggestion that to increase relevance, we should add the descriptions of FBXW7 and the other key players such as Nrf2, CEBPD, and miR223 in prostate cancer. In accordance with the reviewer’s suggestions, we added the descriptions on them in the ‘Introduction’ and ‘Discussion’ sections. (FBXW7: Page 1, line 44-46 and line 48, Ref. [10]; CEBPD: Page 2, line 56-57, Ref. [17]; miR223: Page 2, line 66-67, Ref [21]; Nrf2: Page 10, line 400-402, Ref. [29])

  1. The result that „FBXW7 was significantly transactivated by the treatment with PAX for 12 hr in the LNCaP spheroid model” is not evident from Fig. 1B. Only the mRNA expression (transcription) is increased. Usually, transactivation is a phenomenon occurring with proteins and is independent of mRNA expression. Or do the authors intend to say transcription rather than transactivation? Please clarify and adjust. This mistake occurs frequently throughout the text.

In agreement with the reviewer’s comment, we exchange the term ’transactivation’ for ’transcriptional up-regulation’ or ’up-regulation’. Thank you for your careful reading.

  1. Please indicate in the appropriate Legends that the Western blot results displayed in the Figures are representative of N=4. Also, the Supplement suggests that the replicates be blotted next to each other: please place appropriate legends on the blots in the Supplement for each N.

We greatly appreciate the careful reading by the reviewer. We prepared protein lysates independently from LNCaP spheroids. We add the description ’n = 4’ in the figure legends of the Western blot results (Fig. 1, 6, 8).

  1. P-ERK1/2 and ERK1/2 original blots are missing in the Original Images.

In the previous submission of the images of the original blots, I may have failed to convert the file to PDF. In the re-submission, we confirmed that the PDF file attached as additional data includes all original blots.

  1. It is very suggestive that upon paxilline application ACTB expression is increased, so it is not a suitable housekeeper for this particular study. Recently, ACTB expression has been proposed as a cancer biomarker rather than a „reference gene” as previously used (https://www.ncbi.nlm.nih.gov/pmc/articles/PMC8806805/). Moreover, KCa1.1 interacts with the cytoskeleton so it would be not surprising if paxilline would perturb ACTB expression. Please include another housekeeper, probably one indicative of cell metabolism (e.g., GAPDH) to rule out the possibility that paxilline has effects on the cytoskeletal protein expression.  

When cDNAs were prepared from the same amount of total RNAs, no differences in the Ct values of the reference gene, ACTB, were found between vehicle- and PAX-treated groups. To respond to the reviewer’s doubt that paxilline treatment may affect the expression level of the cytoskeletal protein, ACTB in LNCaP spheroid model, we performed the real-time PCR analysis of the other housekeeping genes, cyclophilin (CYC), β-glucuronidase (GUSB), and glyceraldehyde-3-phosphate dehydrogenase (GAPDH). As shown below, no significant differences in the ACTB expression were found by the treatment with 10 μM PAX for 12 hr in the LNCaP spheroid model.

As a side note, KCa1.1 inhibitor, paxilline is not the focal adhesion phosphoprotein, paxillin that localizes to the cytoskeleton and is responsible for cytoskeleton reconstruction.

human cyclophilin (CYC) (AF022115) (310-419, 120 bp)

forward: 5’-AGGTCCTGGCATCTTGTCCAT-3’

reverse: 5’-TTCACCTTCCCAAGACCACAT-3’

human glyceraldehyde-3-phosphate dehydrogenase (GAPDH) (NM_002046) (595-715, 121 bp)

forward: 5’-GACAACAGCCTCAAGATCATCA-3’

reverse: 5’-TGGCATGGACTGTGGTCAT-3’

human β-glucuronidase (GUSB) (M15182) (1555-1665, 111 bp)

forward: 5’-CTGGAATTTCGCCGACTTCAT-3’

reverse: 5’-CGCAAAATAAAGGCCGAAGTT-3’

  1. On a similar note, is ACTB mRNA expression similarly inhomogeneous as its protein expression? Tools are available to assess whether the „housekeeper” is independent from your experimental conditions (e.g., the geNORM tool). If not, ACTB mRNA would also not be a suitable housekeeper gene.

We found no particular problem in using ACTB as a reference based on the above results. Indeed, in the Western blots using the same amount of proteins, no differences in the ACTB signals were found between vehicle- and PAX-treated groups. We appreciate the reviewer’s valuable and adequate comments.

  1. The claim KCa1.1 inhibition-induced increases in FBXW7 reduced KCa1.1 activity by promoting its protein degradation” is not experimentally assessed in this manuscript, so please show evidence, rephrase, or remove this part. No protein degradation assays have been conducted in this study. One can speculate that the observed effects might be due to protein degradation is the discussion.

We appreciate the reviewer’s careful reading and adequate suggestions. In agreement with the reviewer’s comments, we rephrased it as ‘KCa1.1 inhibition-induced increases in FBXW7 reduced KCa1.1 activity by inhibiting its protein expression’. (Page 10, line 376-377) In addition, we added the description that ’To elucidate whether KCa1.1 inhibition-induced increases in FBXW7 promoted the protein degradation of KCa1.1, further experiments such as protein degradation assay and ubiquitylation detection assay will be needed’. (Page 11, line 435-437)

  1. The next claim (repeated in lines 333-334) is also rather speculation and needs rephrasing and/or removing, or experimental validation: „KCa1.1 inhibition-induced increases in FBXW7 reduced the expression of the core marker of cancer stemness, c-Myc”. Fig. 6 only shows that paxilline impairs c-Myc expression. Here, the consequential link to FBXW7 has not been investigated.

Again, we appreciate the reviewer’s careful reading. In agreement with the reviewer’s comments, we rephrased it as ‘KCa1.1 inhibition reduced the expression of the core marker of cancer stemness, c-Myc, possibly by the transcriptional up-regulation of FBXW7 (Fig. 8). (Page 10, line 379-380)

  1. It is not clear why the Nanog section is not in the main manuscript but in the supplement, even though the authors discuss it heavily in the Discussion. I suggest placing Fig S6 and the corresponding parts into the main text.

In accordance with the review’s comment, the results and descriptions of Nanog experiments in the ‘Discussion’ section were moved to the ‘Results’ section.

  1. Also, regarding lines 333-334, it has not been shown that c-Myc is destabilized—it could simply be less transcribed. Is there any experimental evidence to support increased c-Myc protein degradation, e.g., ubiquitination assay?

As pointed out by the reviewer, we do not have any experimental evidence to support increased c-Myc protein degradation. We amended this sentence as follows. (Page 10, line 379-380)

In the present study, the KCa1.1 inhibition-induced up-regulation of FBXW7 decreased the protein expression level of c-Myc in the LNCaP spheroid model, without changing the expression level of c-Myc transcripts (Fig. 8D-F).

  1. It is not conceivable what the point of the osteosarcoma cell line is in this study. Why use this osteosarcoma cell line specifically? What are the similarities and dissimilarities between prostate cancer and osteosarcoma?  The observed phenomenon that KCa1.1 inhibition increased the transactivation of FBXW7 through the same axis confuses a reader more than it reassures. I suggest removing it as it osteosarcoma is a completely different topic.

In agreement with the reviewer’s suggestion, we removed the results on osteosarcoma MG-63 cells from the ’Discussion’ section and ’Supplementary Figures’.

  1. The Methods section does not describe how spheroid lysates were extracted and how live cells were isolated from the spheroids for confocal imaging.

We appreciate the careful reading by the reviewer. In accordance with the reviewer’s pointing out, the methods of protein lysate preparation and living cell isolation from spheroids were added. [Section 4.2. (Page 12, line 496-497); Section 4.4. (Page 12, line 511-512)]

  1. The siRNA application is also unclear. After siRNA transfection, cells were seeded onto the spheroid-making plates. Were there cells incubated to make spheroids for one week? Please clarify the method.

In accordance with the reviewer’s pointing out, we added the descriptions in Section 4.6. (Page 13, line 549)

  1. Fig. 5C: Please label in color or in the legend which of the data points indicates vehicle and paxilline treatment.

In accordance with the reviewer’s pointing out, we labeled in color (red and blue) in Fig. 5C (Fig. 7C in the new version).

  1. Please include primer sequences in the Supplementary material.

In accordance with the reviewer’s suggestion, we added primer sequence information in Table S1.

  1. DHT abbreviation explanation is missing.

Originally, DHT abbreviation was described in Section 4.1. [5-α-dihydrotestosterone (DHT)].

  1. A minor comment is that the gene expression of human genes and RNA should be written in italic.

Again, we read ‘Instructions for Authors’ in Int J Mol Sci; however, we could not find the reviewer’s indication. We found the description that ‘Italic should only be used where they are required for specific nomenclature (such as species names or journal titles)’.

Round 2

Reviewer 1 Report

Comments and Suggestions for Authors

The authors fully answered the concerns raised in the first revision. 

Reviewer 2 Report

Comments and Suggestions for Authors

The authors answered each comment point by point, and substantially reworked the manuscript. Thereby, the message became much clearer, and the data more transparent.